# *Hymenolepis diminuta* Infection Affects Apoptosis in the Small and Large Intestine

**DOI:** 10.3390/ijerph19159753

**Published:** 2022-08-08

**Authors:** Patrycja Kapczuk, Danuta Kosik-Bogacka, Patrycja Kupnicka, Patrycja Kopytko, Maciej Tarnowski, Agnieszka Kolasa, Dariusz Chlubek, Irena Baranowska-Bosiacka

**Affiliations:** 1Department of Biochemistry and Medical Chemistry, Pomeranian Medical University, Powstańców Wlkp. 72, 70-111 Szczecin, Poland; 2Independent Laboratory of Pharmaceutical Botany, Department of Biology and Medical Parasitology, Pomeranian Medical University in Szczecin, Powstańców Wlkp. 72, 70-111 Szczecin, Poland; 3Department of Physiology, Pomeranian Medical University, Powstańców Wlkp. 72, 70-111 Szczecin, Poland; 4Department of Histology and Embryology, Pomeranian Medical University, Powstańców Wlkp. 72, 70-111 Szczecin, Poland

**Keywords:** apoptosis, hymenolepiasis, parasite–host system, rat

## Abstract

The rat tapeworm *Hymenolepis diminuta* has been shown to cause alterations in gastrointestinal tissues. Since hymenolepiasis induces a number of reactions in the host, it is reasonable to assume that it may also be involved in the mechanisms of apoptosis in the intestines. Individual research tasks included an examination of the effect of *H. diminuta* infection on; (i) the cellular localization of the expression of pro-apoptotic protein Bax and anti-apoptotic protein Bcl-2, as well as caspase-3 and caspase-9, and (ii) the effects of the infection on the expression of Bcl-2, Bax, Cas-3 and Cas-9, at the mRNA and protein levels. Molecular tests (including mRNA (qRT PCR) and the protein (Western blot) expression of Bax, Bcl-2, and caspases-3, -9) and immunohistochemical tests were performed during the experiment. They showed that *H. diminuta* infection activates the intrinsic apoptosis pathway in the small and large intestine of the host. *H. diminuta* infection triggered the apoptosis via the activation of the caspase cascade, including Cas-3 and Cas-9. Hymenolepiasis enhanced apoptosis in the small and large intestine of the host by increasing the expression of the pro-apoptotic gene and protein Bax and by decreasing the expression of the anti-apoptotic gene and protein Bcl-2.

## 1. Introduction

*Hymenolepis diminuta* (rat tapeworm) is a widespread parasite among small rodents and can also occur in humans [1,2]. Despite the fact that the adult tapeworm does not possess clinging organs (hooks) that could damage host tissues, its metabolites can affect the proper functioning of a host’s digestive tract [3]. Due to low pathogenicity and putative immunomodulatory activity, *H. diminuta* can be considered a potential therapeutic agent for the treatment of autoimmune and inflammatory diseases [4,5,6]. To date, the parasite–host system in hymenolepiasis is not yet fully understood in biochemical, histochemical, and molecular terms. It has been shown that hymenolepiasis can cause changes in gastrointestinal tissues. These alterations are mainly related to a decrease in the length of intestinal villi and the deepening of crypts in the digestive system of wild and laboratory rats [7,8].

Metabolites secreted by *H. diminuta* disrupt the host gastrointestinal function, causing increased saliva secretion, inhibition of gastric juice secretion, and increased trypsin activity in duodenal contents [3]. *H. diminuta* infection also causes a decrease in transepithelial electrical potential difference, simultaneously causing a blockade of chloride and potassium ion transport in the rat ileum [9,10]. The presence of *H. diminuta* may also affect smooth muscle contraction in the intestine through the modulation of the muscarinic (M) receptors located in the cell membrane [11] and changes in the composition of intestinal microflora [12,13]. An infection may result in changes in hematological and biochemical blood indicators [14,15]. A study on enzyme activity suggests that hymenolepiasis causes elevated levels of liver enzymes such as alkaline phosphatase and alanine and aspartate aminotransferases [7]. A study by Kosik-Bogacka et al. [10] demonstrated that *H. diminuta* infection causes a significant increase in lipid peroxidation products in the duodenum and jejunum of rats, as well as changes in glutathione levels and antioxidant enzyme activity, such as a decrease in superoxide dismutase activity in various parts of the gastrointestinal tract, a decrease in catalase activity, and an increase in glutathione reductase activity in the rat colon [10].

Numerous studies have shown that, in addition to the aforementioned mechanisms, *H. diminuta* is involved in immune response reactions, such as the expression of Toll-like receptors, and can affect the expression and activity of cyclooxygenases [16,17,18,19,20]. Many excretory and secretory proteins (ESPs) from *H. diminuta* have been identified, which include antigens with potential effects on parasite–host interactions [21,22], yet the signaling of these pathways is still unknown. Further studies are needed to establish the details of molecular mechanisms and immune response pathways during a parasitic invasion. One of the crucial elements is apoptosis—since *H. diminuta* infection affects many reactions in the host, it is conceivable that it may also be involved in the mechanisms of apoptosis in the intestine, the target site of this parasite. 

Apoptosis is programmed cell death controlled by genes and is triggered by both physiological and pathological stimuli. It plays an important role in the prevention of mutations caused by ionizing radiation and reactive oxygen species. Apoptosis also occurs in the course of many diseases such as Alzheimer’s disease, multiple sclerosis, AIDS, type I diabetes, myocardial infarction, stroke, lupus erythematosus, rheumatoid arthritis, ulcerative colitis, cancer, and parasite infections [23,24,25,26,27]. 

The mechanism of apoptosis in intestinal epithelial tissue is a process that maintains homeostasis. It is estimated that the entire process takes about 3–5 days. The loss of contact of the adherent cell with the extracellular matrix results in a form of death called “anoikis”. Cell death signaling can occur through the induction of apoptosis genes as a result of direct DNA damage in the cell under the influence of various stressors, through signals received by tumor necrosis factor receptor (TNFR) family receptors located on the cell surface, cytochrome C, apoptosis-inducing factor (AIF), the activation of cytolytic granules, and reactive oxygen species. The process of apoptosis itself ends up as a result of the overexpression of pro-apoptotic factors, such as Bax; the modulation of anti-apoptotic proteins, such as Bcl-2; and the possibility of the activation of caspase-3 and -9 [28].

The modulation of intestinal epithelial barrier integration is an important mechanism of pathogen invasion, resulting in changes in the host organism. There is a wealth of evidence describing the impairment of intestinal epithelial barrier function by viral, bacterial, and parasitic invasions [29,30,31,32,33,34]. Relationships between organisms living in the same niche, in this case, the intestines, result not only from their direct presence but also from the defense responses of the host itself. It is interesting to note that many parasites have developed a mechanism to modulate host cell apoptosis, creating the possibility of being able to more easily penetrate the intestinal epithelium.

The mechanism of infection effects from *H. diminuta* may be related to changes in expressions at the gene and protein levels responsible for the initiation and progression of apoptosis. Therefore, the aim of our study was to determine whether *H. diminuta* infection can affect apoptosis in the selected structures of the rat intestine.

## 2. Materials and Methods

### 2.1. Choosing a Research Model

Experimental tests on the modulation of apoptosis during *H. diminuta* infection were carried out in Wistar rats. The morphology of the organs of this species is well understood, and it is, therefore, a suitable model organism.

In the planned study, tissues (small and large intestines) from an experiment conducted in 2012–2013 were used, to minimize suffering and the number of animals used. That research was approved by the Local Ethical Committee for Animal Research at the Pomeranian Medical University in Szczecin (Resolution No. 29/2010 dated 7 October 2010, and Annex No. 1/2012 dated 7 November 2012) following Directive 2010/63/EU on the protection of animals used for scientific purposes.

The study was conducted on 30 randomly selected Wistar-strain rats (male) aged approximately 4 months at the start of the experiment. 

The rats (*n* = 30) were divided into five groups:-Control group (*n* = 6)—not infected with *H. diminuta* tapeworm (0 dpi),-Group I (*n* = 6)—8 days after *H. diminuta* infection (8 dpi),-Group II (*n* = 6)—25 days after *H. diminuta* infection (25 dpi),-Group III (*n* = 6)—40 days after *H. diminuta* infection (40 dpi),-Group IV (*n* = 6)—60 days after *H. diminuta* infection (60 dpi).

The *H. diminuta* cysticercoids used to infect the animals had been obtained from cultures from a tapeworm-infected *Tribolium destructor* dark flour beetle (WMS il 1). The cysticercoids dissected from the body cavity of the insects were washed with saline solution and administered to the rats with an intraesophageal probe (5 cysticercoids in 1 mL of saline).

The experiment was always performed at the same time of day, around 9 a.m. The rats used in the experiment received standard food and water *ad libitum*. They were kept in a lit room for 12 h per day. A coproscopic examination of the rats’ feces for parasites was performed before each experiment (3 weeks after infection).

Rats from both the control and experimental groups (groups I–IV) were sacrificed with Tiopentanal (Biochemie GmbH, Viena, Austria) at a dose of 100 mg/kg body weight (b.w.) intraperitoneally (i.p.), then weighed, and blood and organs collected. The small and large intestine sections collected from the animals were cleaned of adhering connective tissue, dissected along the intestinal mesentery, cleared of digestive contents, and divided into pieces approximately 2 cm long. This biological material was frozen at −80 °C.

### 2.2. An Assay Using MicroBCA Protein Assay Kit and Plate Reader

The collected rat intestinal sections (small intestine and large intestine) were prehomogenized by hammering in liquid nitrogen. RIPA lysis buffer (pH 7.4) contained 20 mM Tris; 0.25 mM NaCl; 11 mM EDTA; 0.5% NP-40, 50 mM sodium fluoride, and protease and phosphatase inhibitors (Roche Diagnostics Deutschland GmbH, Mannheim Germany). The whole mixture was incubated for 2 h at 4 °C with constant shaking. After this time, the samples were centrifuged for 20 min at 15,000 *g* (at 4 °C); 30 μL was collected for protein determination, and the resulting supernatant was frozen at −80 °C for further analyses.

For the evaluation of Bcl-2, Bax, Cas-3, and Cas-9 protein expression with the Western blot method, the total protein concentration in the test samples was determined using a MicroBCAPierce™ kit (Thermo Fisher Scientific, Waltham, MA, USA), according to the manufacturer’s instructions, and a plate reader (BiochromAsys UVM 340). This is a kit that uses a colorimetric method to quantitatively measure total protein in a test sample, compared to a protein standard. Using bicinchoninic acid (BCA), a compound that detects Cu^+^ions that are formed when a Cu^2+^ ion is reduced by a protein in an alkaline medium, a purple color is produced. This is the result of the chelation of two BCA molecules with one copper Cu^+^ ion. The colored water-soluble complex has strong absorbance at 562 nm, which also corresponds to the amount of protein in the test sample.

### 2.3. Analysis of Gene Expression in the Small and Large Intestine by Quantitative Real-Time Polymerase Chain Reaction (qRT-PCR)

The quantitative mRNA expression of Bcl-2, Bax, Cas-3, and Cas-9 genes was determined by two-step reverse transcription PCR. The relative expression of the studied genes was determined relative to the average expression of GAPDH, a constitutive expression reference gene (housekeeping genes). Total RNA was isolated from the rat intestinal sections (small and large intestine) using RNeasyMiniKit (Qiagen, Hilden, Germany), according to the manufacturer’s instructions. The concentration and purity of the isolated RNA were determined using a Nanodrop ND-1000 spectrophotometer (Thermo Fisher Scientific™, Waltham, MA, USA). For cDNA synthesis, 1 μg of total RNA isolated from the tissues was used in a 20 μg total sample volume. The resulting template was transcribed into cDNA using a high-capacity reverse transcription kit with Omniscript RT universal primers (random decamers) (Qiagen, Hilden, Germany), according to the manufacturer’s instructions. The quantification of real-time mRNA levels was performed using a 7500 Fast Real-Time PCR System (Applied Biosystems, Foster City, CA, USA) with Power SYBR Green PCR Master Mix reagent (Applied Biosystems, Foster City, CA, USA).

The profile of the reaction carried out was as follows: 95 °C (15 s), 40 cycles at 95 °C (15 s), and 60 °C (1 min). The primer sequences were designed to span an intron between a sense primer and an antisense primer to prevent the multiplication of genomic DNA. Each sample was analyzed in duplicate (in two technical replicates), and mean Ct values were used in further studies. An analysis of these relative changes in gene expression between samples was performed using the 2^−ΔΔCT^ method [35] Each reaction ended with a melt curve analysis. The following primer pairs were used (Table 1):

### 2.4. Analysis of Protein Expression in the Small and Large Intestine by Western Blot- Measurement of Bcl-2, Bax, Cas-3, and Cas-9 Protein Expression

Protein electrophoresis was performed using 11% polyacrylamide gel electrophoresis (SDS-PAGE), with 30 μg protein/well for separation. A wet transfer method was then applied, i.e., fractionated proteins were transferred onto a 0.2 µm PVDF membrane (Thermo Fisher Scientific™, Waltham, MA, USA) at 75 V for 60 min. Subsequently, the membrane was blocked with 3% bovine serum albumin (BSA) in blocking buffer for 1 h at room temperature. The expression of apoptotic proteins was determined by immunodetection with specific antibodies. Accordingly, I-primary monoclonal antibodies against Bcl-2 (cat. no: sc-23960, Santa Cruz Biotechnology, Dallas, TX, USA), Cas-3 (cat. no: sc-56053, Santa Cruz Biotechnology, Dallas, TX, USA), Cas-9 (cat.: sc-56076, Santa Cruz Biotechnology, Dallas, TX, USA) and polyclonal against Bax (cat. no: sc-7480, Santa Cruz Biotechnology, Dallas, TX, USA) at a dilution of 1:400 overnight at refrigerator temperature. Then, it was incubated with a secondary antimouse antibody at a dilution of 1:4000 for one hour at room temperature. Beta actin (Cat. No: sc-47778, Santa Cruz Biotechnology, USA) was used as the reference protein. To visualize protein expression, an ECL Advance Western Blotting Detection Kit (GE Healthcare, Chicago, IL, USA) was used for chemiluminescence visualization, and the bands were developed using MolecularImagerChemiDock XRS+ (Bio-Rad, Hercules, CA, USA).

### 2.5. Immunohistochemical Analysis of Bax, BCl-2, Cas-3, and Cas-9 in the Small and Large Intestine

Paraffin-embedded sections (3–5 μm) of the small and large intestines of the control and *H. diminuta*-infected rats were subjected to immunohistochemical staining to visualize the apoptotic proteins Bcl-2, Bax, Cas-3, and Cas-9.

The tissues were placed in a 4% buffered solution of formalin for 24 h, then a dehydration series was performed, and the tissues were embedded in paraffin for 3 h. The blocks were sliced into 3–5 µm thick sections using a microtome (HM 340E Electronic Rotary Microtome, Thermo Fisher Scientific™, Waltham, MA, USA). The sections were then spotted onto histology slides ((3-Aminopropyl)triethoxysilane, Cat. No. J2800AMNZ, Thermo Fisher Scientific™, Waltham, MA, USA), which were deparaffinized and rehydrated. To reveal antigenic determinants, deparaffinization was performed by microwave cooking twice (700 W, 4 and 3 min) in 10 nM citrate buffer (pH 6.0). Immunohistochemistry was performed using overnight incubation at 4 °C with specific primary monoclonal antibodies against Bcl-2 (sc-23960, Santa Cruz Biotechnology, Dallas, TX, USA), Cas-3 (sc-56053, Santa Cruz Biotechnology, Dallas, TX, USA), Cas-9 (cat. no.: sc-56076, Santa Cruz Biotechnology, Dallas, TX, USA), and polyclonal against Bax (cat.: sc-7480, Santa Cruz Biotechnology, Dallas, TX, USA) at a dilution of 1:100. The sections were then stained with an avidin–biotin–peroxidase system with diaminobenzidine as the chromogen. For this purpose, a commercial Dako LSAB + System-HRP kit (Dako Inc., Canpinteria, CA, USA) was used according to the manufacturer’s instructions. Brown staining was considered a positive reaction indicating the presence of the antigen, and negative control was performed without primary antibodies. Additionally, hematoxylin contrast staining was used to visualize the cytoplasm and cellular nuclei. A light microscope (Leica DM5000 B, Wetzlar, Germany) integrated with a camera was used to visualize the slides. 

On 6 microphotographs (obj. mag. ×20) from each group (8, 25, 40, 60 dpi), immunopositive and immunonegative epithelial cells were counted. The calculations were made independently by two histologists, and then the obtained results were averaged. The following scale has been assigned: negative (−) when immunopositive cells constituted from 0 to 5%, very weakly positive (+/−) from 6% to 20%, weakly positive (+) from 21% to 49%, moderately positive (++) from 50% to 75%, or strongly positive (+++) from 76% to 100%.

### 2.6. Statistical Analysis

Analysis of the obtained results was performed using Statistica 13.0 software (StstSoft Polska sp. z o.o., Kraków, Poland). The arithmetic mean ± SD was calculated for each of the studied parameters. The distribution of results for each variable was obtained using a Shapiro–Wilk test. As most of the obtained results differed from a normal distribution, in further analyses, a non-parametric Kruskal–Wallis test was used to indicate differences between the study groups. Probabilities at *p* ≤ 0.05 were considered statistically significant.

## 3. Results

### 3.1. Changes in Apoptosis Gene Expression in the Small and Large Intestines of the Rat during H. diminuta Infection

The expression of Bcl-2, Bax, Cas-3, and Cas-9 genes in the small and large intestine of the control and *H. diminuta* infected rats was determined at the mRNA level. The expression of Bax, Cas-3, and Cas-9 genes was lower in the uninfected rats compared to the infected rats. In contrast, an elevated Bcl-2 gene expression in the control group was observed in both the small and large intestines compared to the other groups (Figure 1).

In the small intestine of the *H. diminuta*-infected rats, a statistically significant increase in Bax gene expression was observed at 8 dpi (*p* < 0.01), 25 dpi (*p* < 0.01), and 40 dpi (*p* < 0.05). At 60 dpi, the change in the Bax gene expression level was significantly downregulated; however, this was not statistically significant (Figure 1A). Similar results were obtained by analyzing the expression of Cas-3 and Cas-9 genes in the small intestine. In the case of Cas-3 gene, at 8 dpi (*p* < 0.01), 25 dpi (*p* < 0.05), 40 dpi (*p* < 0.01), and 60 dpi (*p* < 0.01), the expression was statistically significantly higher and higher on the day of infection, except for 60 dpi (Figure 1B). In contrast, Cas-9 gene expression at 8 dpi (*p* < 0.01), 25 dpi (*p* < 0.01), and 40 dpi (*p* < 0.05) was statistically significantly higher compared to the control group but not at 60 dpi and 0 dpi (Figure 1C). Bcl-2 gene expression in the small intestine was statistically significantly much lower at 8, 25, 40, and 60 dpi than at 0 dpi (8: 3×, 25: 3×, 40: 2×, and 60: 2×, respectively) (*p* > 0.01) (Figure 1D).

In the colon, Bax gene expression was slightly elevated at 8 and 40 dpi compared to the control group (not statistically significant). At 60 dpi, the expression of Bax was significantly downregulated (*p* < 0.01). The upregulation of the Cas-3 gene was observed at 8 and 40 dpi; however the changes were not statistically significant. At 25 and 60 dpi, the expression of Cas-3 was downregulated (not significant). In the case of Cas-9, at 8 dpi (*p* < 0.05), 25 dpi (*p* < 0.01), 40 dpi (*p* < 0.01), and 60 dpi (*p* < 0.05), the expression was higher in comparison to the control group (8: 4×, 25: 10×, 40: 16×, 60: 24×, respectively). Bcl-2 gene expression in the colon decreased post infection and reached a significantly lower level at 60 dpi (*p* < 0.01) (Figure 2).

### 3.2. Changes in Apoptotic Protein Expression in the Small and Large Intestine of the Rat during H. diminuta Infection

*H. diminuta* infection affected the expression of apoptotic proteins in both the small and large intestine. The expression of Bax protein in the intestines of the infected rats was significantly higher than in the control group. In the small intestine, Bax protein expression increased the most at 8 dpi (122%) (*p* < 0.01) and less on the other days: 25 dpi (50%) (*p* < 0.01), 40 dpi (13%), and 60 dpi (80%) (*p* < 0.01) compared to 0 dpi. In the colon, there was an increase of Bax expression at 8 dpi (28%) (not significant). At 25 dpi (53%), 40 dpi (70%), and 60 dpi (77%) the level of Bax protein decreased significantly (*p* < 0.05) compared to the 0 dpi group (Figure 3).

The expression of anti-apoptotic protein Bcl-2 in the small intestine varied according to the duration of hymenolepiasis. Protein expression decreased significantly (*p* < 0.05), by 32% at 8 dpi, and 70% at 25 dpi. On subsequent days, expression was higher vs. the control. At 40 dpi, expression increased by 13% (not significant) and at 60 dpi by 44% (*p* < 0.05). In the colon of the infected rats, Bcl-2 protein expression was significantly higher at 8 dpi (41%), 25 dpi (43%), and 60 dpi (50%) compared to the uninfected rats (*p* < 0.01) (Figure 4).

In the small intestine, caspase-3 protein expression was the highest at 40 dpi, and it was 20% higher compared to the control group (*p* < 0.05). On subsequent days, expression was lower vs. the control. In the colon, there was a decrease of caspase-3 expression at 8 dpi (43%) (not significant). At 25 dpi, expression decreased by 41%, and at 60 dpi, it decreased by 66% (*p* < 0.05). In contrast, in the colon, caspase-3 expression was the highest in the uninfected rats, and at 8 (66%) and 25 (73%) days post-infection, the lowest expression of caspase-3 was observed (*p* < 0.05), increasing at 40 (64%) and 60 (56%) days, but was still significantly lower in the uninfected rat group (*p* < 0.05) (Figure 5).

For caspase-9 expression in the small intestine, the highest level was observed at 60 dpi (*p* < 0.05) (148%). From 8 dpi, expression was lower vs. the control. At 8 dpi, expression decreased 17%, at 25 dpi 26%, and 40 dpi 22% (not significant). Caspase-9 expression in the colon at 8 dpi was lower (42%) compared to the control group; however, the change was not significant. The level of Cas-9 protein in the colon significantly increased at 25 dpi (139%) (*p* < 0.05) vs. the control, and, in subsequent groups, it decreased, reaching a significantly lower level at 60 dpi (66%). At 40 dpi, expression increased by 2% (not significant) (Figure 6).

### 3.3. Immunohistochemical Analysis of Bax, Bcl-2, Cas-3, and Cas-9

The results of immunohistochemical reactions showed that *H. diminuta*-infected rats (at 8, 25, 40, and 60 dpi) showed increased Bax, Cas-3, and Cas-9 immunoexpression and decreased Bcl-2 in small intestinal epithelial and colonic epithelial cells compared with the uninfected controls (Table 2).

Figure 7 shows Bax immunoexpression in the small intestine of the control animals (0 dpi) and on consecutive days (8 dpi, 25 dpi, 40 dpi, 60 dpi) in the *H. diminuta*-infected animals. In the small intestinal wall of the control animals (0 dpi) (Figure 7A,a), the IHC reaction was lowest and was marked in single epithelial cells of the villi and intestinal crypts (red arrows) and in a few cells in the connective tissue lining of the intestinal wall mucosa (blue arrows). By far the strongest immunohistochemical reaction showing Bax expression (Figure 7B,b) was observed in epithelial cells (red arrows), and the number of immunopositive cells was higher than in control rats (8 dpi). Bax-positive cells were also present in the lamina propria of the mucosa of these animals (blue arrows). On subsequent days after *H. diminuta* infection, the pattern of Bax expression was similar, but the intensity became lower and lower (Figure 7C,c,D,d; red and blue arrows) before elevating again by 60 dpi (Figure 7E,e; red and blue arrows). In animals of the 25 dpi group, a few Bax-positive intraepithelial lymphocytes were observed in the epithelium (Figure 7C; green arrows).

In contrast, in the intestinal wall of the control animals, Bax immunoexpression was very low, and only a few cells of the lining epithelium, mainly cup cells, were Bax-positive (Figure 8A,a; red arrows). A similar level of IHC reaction was observed in the 40 dpi group (Figure 8D,d; red arrows) but not in the intestinal crypts. In contrast, at 8, 25, and 60 dpi, the reaction was higher and present mainly in enterocytes (Figure 8B,b,C,c; red arrows) and cup cells (Figure 8E,e; red arrows). In the lamina propria of the mucosa, Bax-positive cells were present during the early days of hymenolepiasis (Figure 8B,b,C,c; blue arrows).

The immunoexpression of Bcl-2 in the intestinal wall of animals of all of the infected groups was quite low, while in the control group (Figure 9A,a) and at 8 dpi (Figure 9B,b), they were quite comparable, mildly marked in epithelial cells (red arrows), and somewhat stronger in cells of interstitial tissue (blue arrows). At 25 dpi (Figure 9C,c), the IHC reaction was very weak and barely marked in epithelial cells (red arrows) was and slightly stronger in cells of the mucosa lamina propria (blue arrows). By 40 dpi and 60 dpi, the level of Bcl-2 immunoexpression had increased, and these groups (Figure 9D,d,E,e) were at comparable levels, visible just in some epithelial cells of the villi, intestinal crypts (red arrows), and cells (blue arrows) of the lamina propria of the mucosa.

The highest Bcl-2 immunoexpression was observed in the large intestine of the control animals (Figure 10A,a), primarily in the epithelial (cup) cells of intestinal crypts (red arrows), whereas in the lamina propria of the mucosa, Bcl-2-positive cells (blue arrows) were scarce. At 8 dpi, 25 dpi, 40 dpi, and 60 dpi, the level of Bcl-2 immunoexpression was comparable to 0 dpi (Figure 10B,b,C,c,D,d,E,e) and was mainly seen in the glandular epithelium of the intestinal crypts (red arrows).

In the control group (Figure 11A,a), the level of caspase-3 immunoexpression in the small intestine was lowest, noted in the cytoplasm and nucleus of some epithelial cells, mainly intestinal villi (red arrows). Furthermore, a few cells of the lamina propria of the submucosa were immunopositive (blue arrows). From 8 dpi with *H. diminuta* to 40 dpi (Figure 11B,b,C,c,D,d), the level of caspase-3 immunoexpression gradually increased and was characterized by similar localization, that is, in epithelial and interstitial cells (red and blue arrows, respectively), except that the latter were very abundant in the mucosa of animals of the 8 dpi group (Figure 11B,b, blue arrows). The level of caspase-3 immunoexpression in the small intestinal epithelium of the 60 dpi group rats (Figure 11E,e) was similar to that of the 8 dpi group.

In the control animals (Figure 12A,a), the level of caspase-3 immunoexpression in the colon was relatively high, noticeably in epithelial cells (red arrows) and connective tissue cells of the mucosal lamina propria (blue arrows). On successive days after *H. diminuta* infection (8 dpi, 25 dpi, 40 dpi, 60 dpi), the level of caspase-3 immunoexpression fluctuated: it increased by 8 dpi (Figure 12B,b), decreased by 25 dpi (Figure 12C,c), increased by 40 dpi (Figure 12D,d), and again decreased by 60 dpi (Figure 12E,e). The expression pattern of caspase-3 in the colon wall was similar to the control animals, i.e., it was seen in the epithelium and the mucosal connective tissue (red, blue arrows).

In the small intestine of the control animals (Figure 13A,a), the level of caspase-9 immunoexpression was low and seen in the cytoplasm or nucleus of the epithelial cells (red arrows) and the mucosal lamina propria cells (blue arrows). By 8 dpi (Figure 13B,b), the level of caspase-9 immunoexpression increased and was seen in the cytoplasm of most epithelial cells (red arrows) and in the interstitial cells (blue arrows). By 25 dpi and 40 dpi, the intensity of immunoreactivity for caspase-9 gradually decreased, with the localization remaining constant (Figure 13C,c,D,d, red, blue arrows). However by 60 dpi (Figure 13E,e, red, blue arrows), the immunoexpression was comparable to the control group (Figure 13A,a).

In the control animals (0 dpi), caspase-9 immunoexpression in the colon was at a very low level, with few cells of the epithelium and interstitial tissue being positive for this marker of apoptosis (Figure 14A,a, red, blue arrows). As time passed after *H. diminuta* infection (Figure 14B,b,C,c,d,d,E,e), the level of caspase-9 immunoexpression increased linearly. More and more epithelial cells (surface and intestinal crypts) and connective tissues showed cytoplasmic and nuclear immunohistochemical reactions (red, blue arrows).

## 4. Discussion

The presence of *H. diminuta* in the intestine of the host affects the function of the organ and its immunity. Numerous studies indicate the interference of *H. diminuta* with the host cell response, with cellular signals, and with the invasion mechanism itself. In our study, we demonstrated the modulation of small and large intestine apoptosis mechanisms during *H. diminuta* infection by analyzing the expression of specific apoptotic genes and proteins: Bax, Bcl-2, Cas-3, and Cas-9, using quantitative real-time polymerase chain reaction (qRT-PCR), Western blot, and immunohistochemistry (IHC).

It is interesting to note that many parasites have developed a mechanism to modulate host cell apoptosis, thus creating the possibility of easier penetration into the intestinal epithelium. A recent study showed that *Giardia* sp. can induce apoptosis in intestinal epithelial cells by activating both the intrinsic and extrinsic pathways [36]. Numerous reports suggest that, depending on the strain, this parasite can destructively affect the host intestinal epithelium by inducing different mechanisms. Similar observations have been made for *H. diminuta*, which causes histopathological changes in the intestines, as well as altering molecular and immunological mechanisms [37]. 

The present study demonstrates the effect of *H. diminuta* infection on the apoptosis of host intestinal epithelial cells in a rat model. Molecular and histological studies were performed to determine the induction of apoptotic mechanisms in the small and large intestine and indicated that *H. diminuta* can stimulate enterocyte apoptosis through the activation of caspase-3 and caspase-9; changes in protein expression, including the high expression of pro-apoptotic proteins (Bax); and the low expression of anti-apoptotic proteins (Bcl-2). 

The Bcl-2 family proteins regulate the intrinsic apoptosis pathway centered in mitochondria. The ratios of pro-apoptotic Bax and anti-apoptotic Bcl-2 are correlated with the release of cytochrome C and the subsequent activation of the caspase cascade [38]. Literature data indicates that caspase-3 is the main effector involved in apoptosis, and its activation directs the irreversible step of apoptosis. The release of cytochrome c in mitochondria is one of the initial diagnostic features of nuclear cell apoptosis [39]. Bcl-2 has an anti-apoptotic effect by reducing the release of cytochrome c from mitochondria to inhibit the activation of caspase-3. In contrast, Bax protein is a key component of ion channels located in the mitochondrial membrane. Moreover, it induces cytochrome c translocation across mitochondrial membranes and the formation of apoptotic bodies. Bax activates caspase-9 and caspase-3, which consequently leads to programmed cell death [40]. Therefore, in the present study, the molecular mechanisms associated with such effects are investigated by analyzing the stimulation of hymenolepiasis on the expression levels of cleaved (activated) caspase-3 and caspase-9 and mitochondria-associated apoptotic proteins, Bcl-2 and Bax. In the present study, their expression was checked at both the mRNA and protein levels. 

The hypothesis of apoptosis induction by *H. diminuta* under experimental conditions was confirmed by an observed increase in the expression of the pro-apoptotic protein Bax. Moreover, the theory was supported by immunohistochemical results in which its localization was determined. The study indicates an increased presence of Bax in epithelial cells, lamina propria of the small intestinal mucosa, and enterocytes of cup cells and lamina propria of the colonic mucosa. In addition, Bax-positive intraepithelial lymphocytes were found in the small intestine during hymenolepiasis. In contrast, Bcl-2 expression was significantly lower in the different *H. diminuta*-infected groups at both the mRNA and protein levels. Similar results were obtained for immunohistochemical analysis. Moreover, the increase in Bax protein shifted the Bax/Bcl-2 ratio in favor of apoptosis. Additionally, it was shown that caspase-3 and caspase-9 were noticeably activated. Caspase-3 and caspase-9 mRNA and protein expressions were significantly higher in the small and large intestine during *H. diminuta* infection compared to the uninfected control group. The IHC results were analogous to the Western blot and qRT-PCR results and suggest that *H. diminuta*-induced cell apoptosis was associated with the activation of a mitochondria-related pathway in the rat intestine.

Similar results apply to infections with other intestinal parasites. Numerous studies show that *Giardia* sp. affects the host similarly to *H. diminuta*, except that it also inactivates the PARP protein [32,41,42,43]. Chin et al. (2002) found that *G. lamblia* trophozoites can activate the process of apoptosis in non-transformed human small intestinal epithelial cells, depending on the parasite strain. Furthermore, *G. lamblia*, by inducing apoptosis, reduced the integrity of the host intestinal epithelial barrier. In addition, it was shown that these changes may be dependent on caspase-3 [41], whose increased activity was also observed during *H. diminuta* infection. Panaro et al. (2007) conducted a study on apoptosis in a parasite–host system using human ileal adenocarcinoma line cells (HCT-8). The interaction of colon epithelial cells with *G. intestinalis* trophozoites showed that the parasite can modulate apoptosis processes in the host intestines through DNA fragmentation, caspase-3 stimulation, PARP degradation, and the regulation of protein expression: Bcl-2 and Bax [36]. An experiment by Troeger et al. (2007) showed only a slight effect of *G. lamblia* on the apoptosis process among patients infected with *G. lamblia*, where it increased by 0.2% compared to the control group [43]. In contrast, Roxstrom-Lindquist et al. (2005) suggest that this parasite not only can induce apoptosis but also induce large changes in cell gene expression, including pro-inflammatory factors in intestinal epithelial cells [42]. These reports indicate that the adaptive mechanisms involving the production of surface proteins and the secretion of secretory products may be responsible for the induction of apoptosis glideos [37]. 

Analyzing the results obtained, it can be assumed that both the mere presence of the parasite in the intestines and its ES may indicate the pro-apoptotic potential of *H. diminuta* and the putative function of caspase-dependent apoptosis in the pathogenesis of hymenolepiasis. Given the location of the *H. diminuta* and its impact on normal intestinal function, studies of the modulation of apoptotic pathways in the intestines in the parasite–host system are still a point of interest. *H. diminuta* is known to induce mucosal barrier changes during infection [7,10].

A recent study by Hayes et al. (2017) demonstrates that cell proliferation and apoptosis both increase in the colon during chronic infection in mice with *Trichuris muris*. A significant increase in the number of cells indicative of apoptosis was observed at the base of the crypts, while in more infected animals, successively more proliferating cells were found further down the intestinal crypt. Interestingly, no effect of infection on proliferation or apoptosis was observed in the small intestine, despite the presence of morphological changes at this site [44]. In contrast, in the present study, the immunoexpression of the pro-apoptotic protein Bax and caspase-3 and caspase-9 were significantly higher in both the small intestine (the site of the parasite) and the large intestine, indicating increased cell turnover in these structures. Although the target site of *H. diminuta* is the small intestine, the observed changes in the large intestine may suggest the secretion of substances into the intestinal lumen, simultaneously inducing apoptotic processes in later parts of the gastrointestinal tract.

Target cell death is associated with the contact-dependent caspase activation of infected cells [45]. In the present study, the expression of caspase-3, as well as caspase-9, was elevated in both mRNA and protein levels compared to the control group. Thus, the extracellular parasite can induce apoptosis in cells independently of the membrane death receptors, i.e., the external apoptotic pathway and of the internal apoptotic pathway. Thus, it can induce apoptosis through direct contact upon attachment to the intestinal epithelium. Similar observations apply to *Entamoeba histolytica* infections. The cytotoxicity of *E. histolytica* has also been shown to depend on effector proteins, which include amoebapore and proteases responsible for parasite invasion and host cell apoptosis. In addition, these proteins are thought to be responsible for the direct activation of caspase-3 [46].

In hymenolepiasis, changes in intestinal morphology are observed, including atrophy of intestinal villi [8]. Similar observations have been noted in *Nippostrongylus brasiliensis* infection. A study by Hyoh et al. (1999) showed that this parasite can cause partial villous atrophy and crypt proliferation in the small intestinal mucosa. Furthermore, it was shown that intestinal villi atrophy was associated with increased apoptosis and a loss of adhesion in host epithelial cells [47]. Hence, it can be speculated that *H. diminuta* infection inducing analogously similar changes in intestinal structures may induce the same apoptotic mechanism as *N. brasiliensis*. Thus, the induction of the apoptosis pathway in intestinal epithelial cells may be linked to the host defense mechanism of rapidly removing damaged cells. It is supposed that the apoptosis of villi cells can be induced directly by molecules from the parasite itself, as well as from ES [48,49].

In an experiment by Kuroda et al. (2002), the effects of *N. brasiliensis* and ES on the intestinal epithelial cell line IEC-6 were examined. The study indicates that *N. brasiliensis* secretes biologically active molecules that can induce apoptosis in host intestinal epithelial cells along with the upregulation of Fas expression. Although the mechanism of apoptosis induction itself remains undocumented, exposure to ES has been shown to trigger apoptosis mechanisms in cells, including nuclear fragmentation, caspase-3 activation, and specific PARP cleavage [50].

*H. diminuta*, similarly to the other mentioned intestinal parasites, may contribute not only to single-cell disorders but also to the functioning of the whole organ, especially as the parasites have evolved complex mechanisms to modulate parasite–host interactions. It has been observed that *Cryptosporidium*, depending on the species and the particular stages of the developmental cycle, also affects the mechanisms of host apoptosis pathways [51,52,53,54,55,56,57].

Our study demonstrated that *H. diminuta* can induce apoptosis in the host intestines. A significant increase in the expression of caspase-3 and caspase-9, which is correlated with the activation of the caspase cascade during apoptosis, indicates that *H. diminuta* induces the intrinsic pathway of apoptosis at the parasite target site [34,35]. Our data show that apoptosis is only one of many mechanisms occurring during intestinal parasite infections and that the parameters determined in this study may be useful markers of apoptosis in parasitic diseases. Many questions remain unanswered, such as whether increased induction of physiological turnover of intestinal epithelial cells is stimulated at the immune level to remove parasites by causing their eventual expulsion along with their immediate environment, the intestinal epithelium; how the inhibition of apoptotic pathways affects the host intestinal epithelium; and whether anti-apoptotic treatment during infection with these parasites is possible without harm to other tissues. 

The modulation of host apoptosis under parasitic invasion is not yet fully understood and requires further study on other intestinal parasites. It opens an interesting new area of research due to the wide range of subject matter and the increasing number of infections with “new” and opportunistic intestinal parasites. Furthermore, since there is no clear and publicly available animal model for intestinal parasite infections, future experiments should also focus on the development of animal models to support in vitro studies with data obtained in the parasite–host system.

## 5. Conclusions

Infection with *H. diminuta:* (i) activates the intrinsic apoptotic pathway in the small and large intestine of the host; (ii) triggers apoptosis via the activation of a caspase cascade, such as with caspase-3 and caspase-9; (iii) enhances apoptosis in the small and large intestine of the host by increasing the expression of the gene and pro-apoptotic protein Bax and by decreasing the gene and protein expression of anti-apoptotic Bcl-2.

Programmed cell death is an important process for both parasite survival and pathogenicity and for the host to maintain defense mechanisms.

## Figures and Tables

**Figure 1 ijerph-19-09753-f001:**
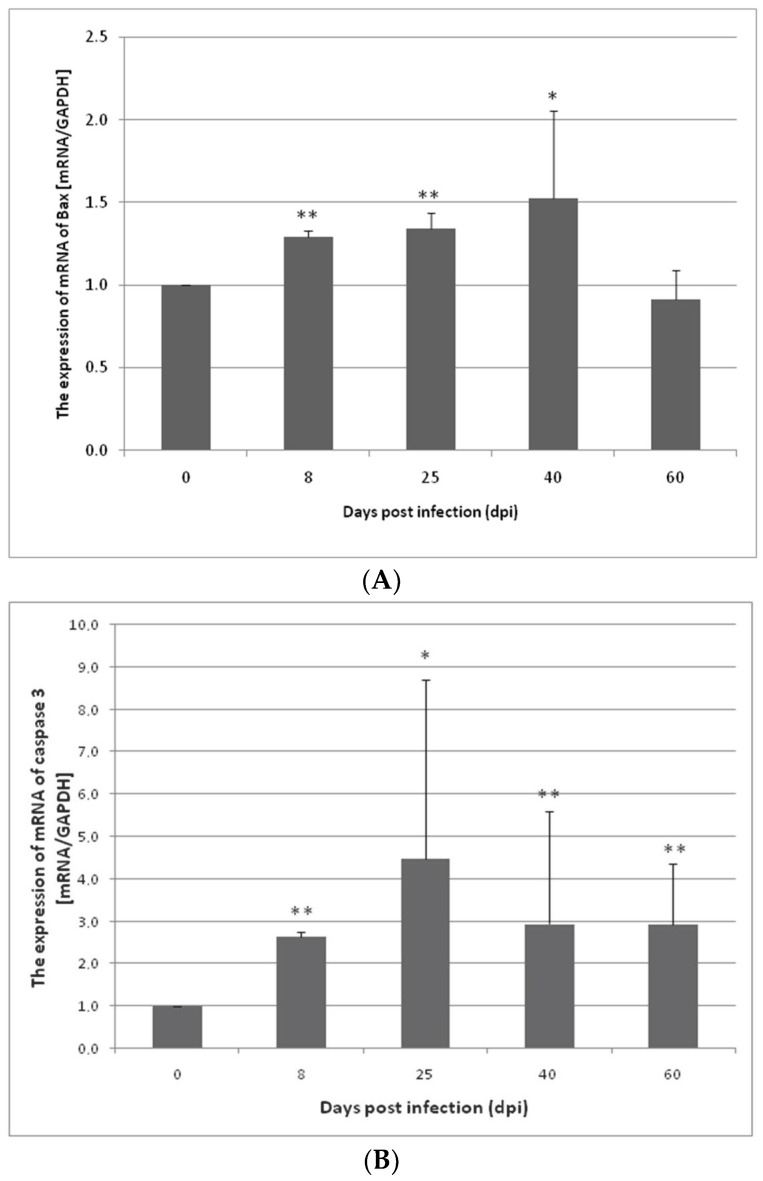
Expression of Bax (**A**), caspase-3 (**B**), caspase-9 (**C**), and Bcl-2 (**D**) genes at the mRNA level in the small intestine from uninfected and *H. diminuta*-infected rats. Data represent the arithmetic mean ± SD and are representative of individual groups of six animals in each experiment. Statistical analysis was performed using the Kruskal–Wallis test, * *p* < 0.05, ** *p* < 0.01 vs. 0 dpi.

**Figure 2 ijerph-19-09753-f002:**
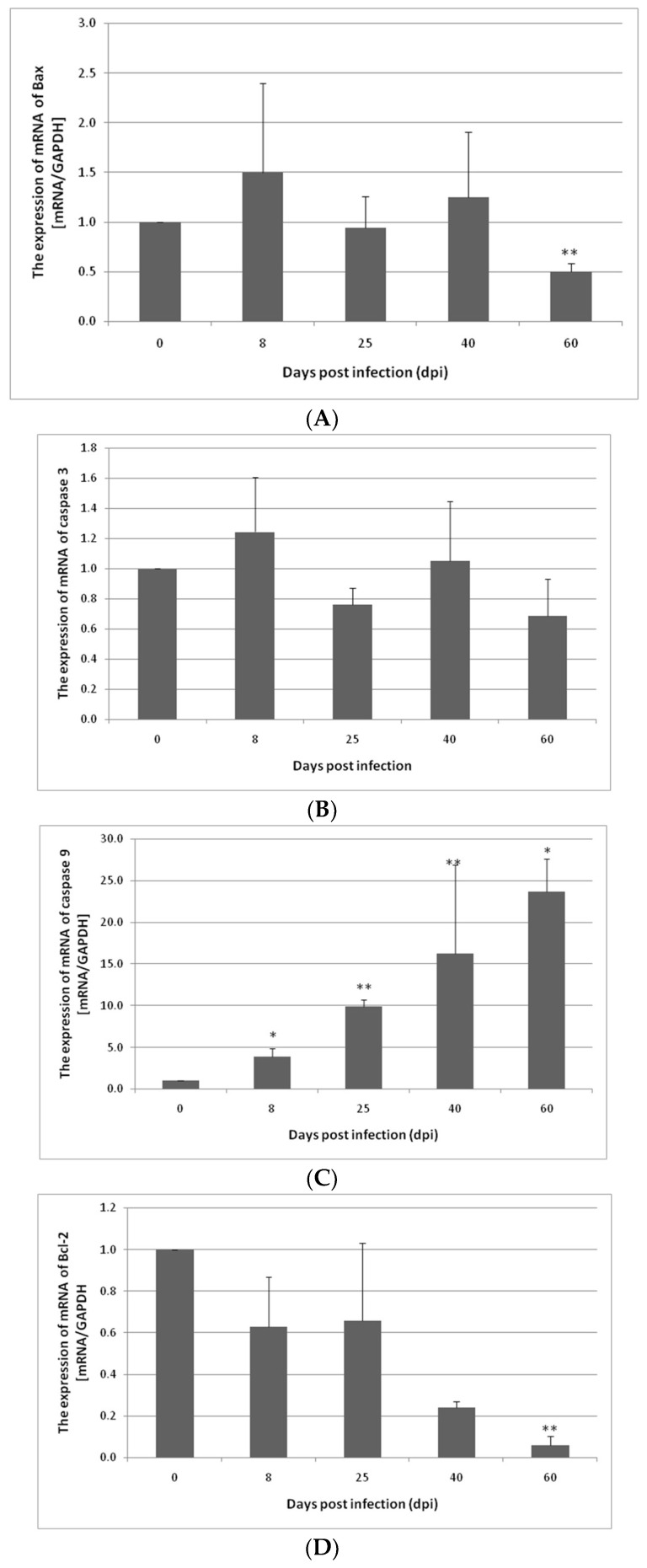
Expression of Bax (**A**), caspase-3 (**B**), caspase-9 (**C**), and Bcl-2 (**D**) genes at the mRNA level in the colon from uninfected and *H. diminuta*-infected rats. Data represent the arithmetic mean ± SD and are representative of individual groups of six animals in each experiment. Statistical analysis was performed using the Kruskal–Wallis test, * *p* < 0.05, ** *p* < 0.01 vs. 0 dpi.

**Figure 3 ijerph-19-09753-f003:**
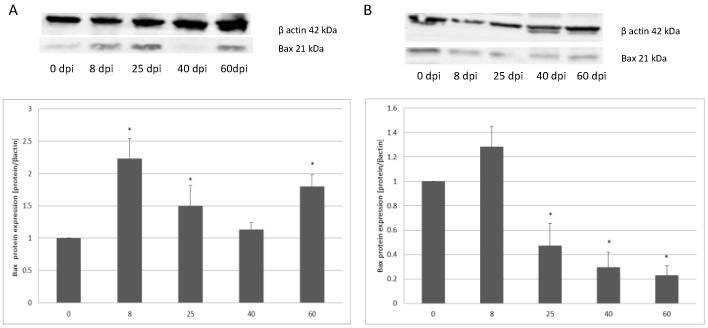
Representative Western blot bands and densitometric analysis of Bax protein level in the small intestine (**A**) and large intestine (**B**) from uninfected and *H. diminuta*-infected rats. Data represent the arithmetic mean ± SD and are representative of individual groups of six animals in each experiment. Statistical analysis was performed using the Kruskal–Wallis test, * *p* < 0.05, vs. 0 dpi.

**Figure 4 ijerph-19-09753-f004:**
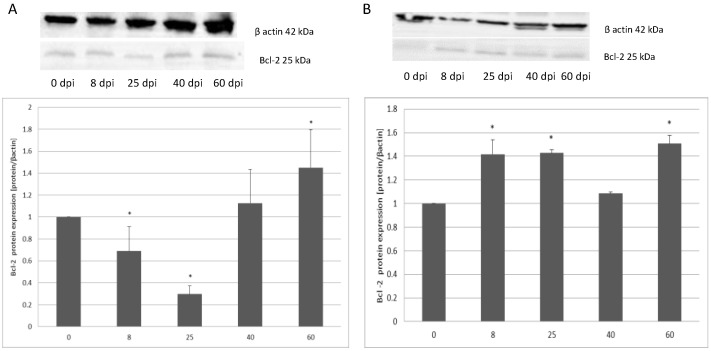
Representative Western blot bands and densitometric analysis of the Bcl-2 protein level in the small intestine (**A**) and large intestine (**B**) from uninfected and *H. diminuta*-infected rats. Data represent the arithmetic mean ± SD and are representative of individual groups of six animals in each experiment. Statistical analysis was performed using the Kruskal–Wallis test, * *p* < 0.05, vs. 0 dpi.

**Figure 5 ijerph-19-09753-f005:**
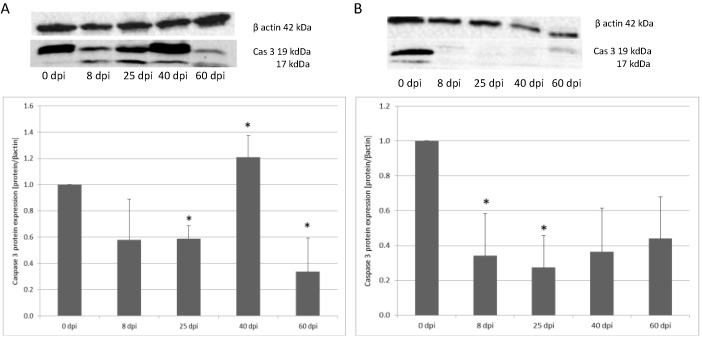
Representative Western blot bands and densitometric analysis of the caspase-3 protein level in the small intestine (**A**) and large intestine (**B**) from uninfected and *H. diminuta*-infected rats. Data represent the arithmetic mean ± SD and are representative of individual groups of six animals in each experiment. Statistical analysis was performed using the Kruskal–Wallis test, * *p* < 0.05 vs. 0 dpi.

**Figure 6 ijerph-19-09753-f006:**
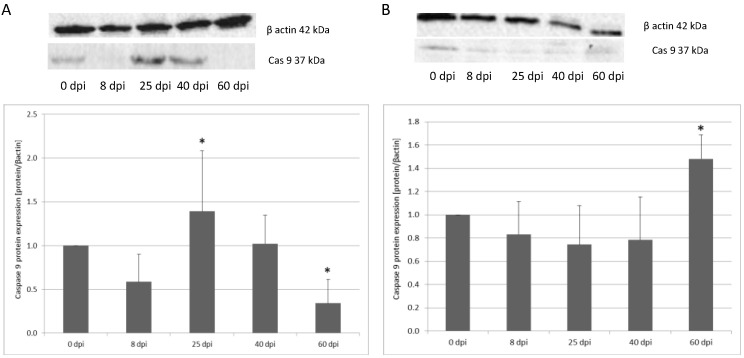
Representative Western blot bands and densitometric analysis of the caspase-9 protein level in the small intestine (**A**) and large intestine (**B**) from uninfected and *H. diminuta*-infected rats. Data represent the arithmetic mean ± SD and are representative of individual groups of six animals in each experiment. Statistical analysis was performed using the Kruskal–Wallis test, * *p* < 0.05 vs. 0 dpi.

**Figure 7 ijerph-19-09753-f007:**
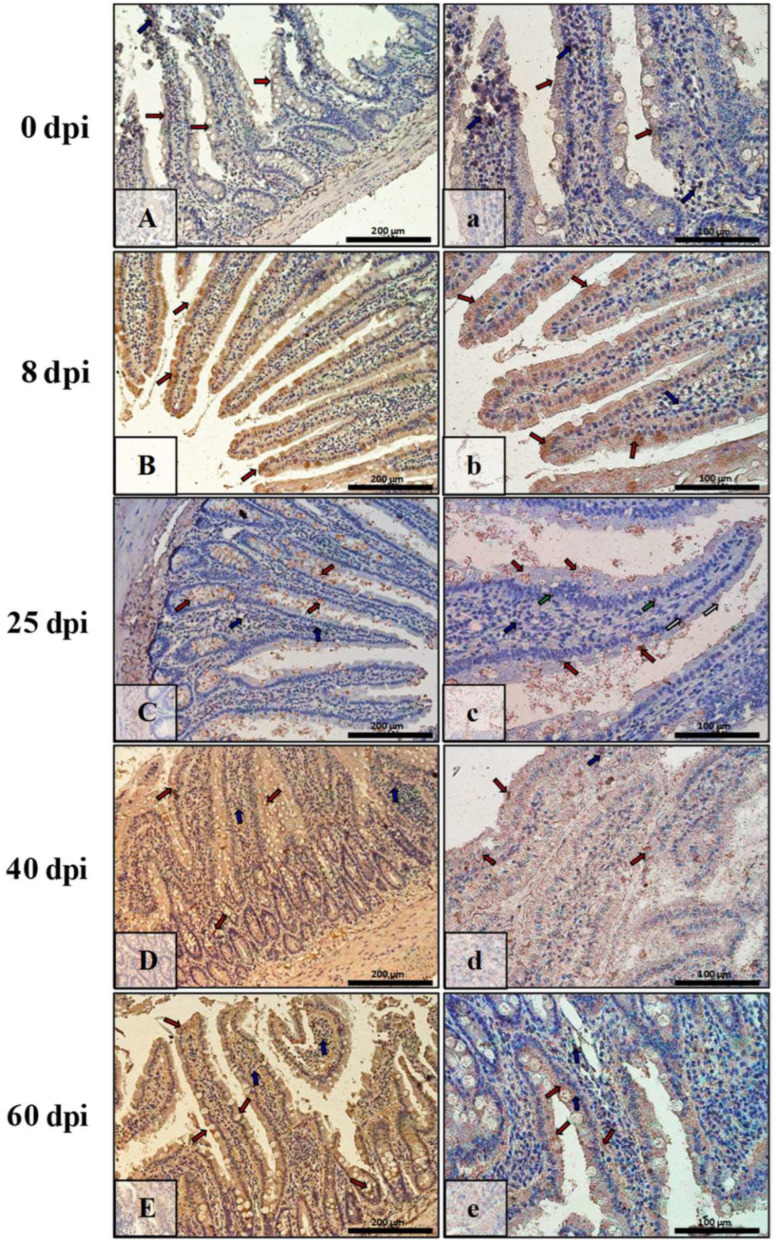
Representative microphotographs showing Bax immunoexpression in the small intestine of control animals (**A**,**a**) and on subsequent days (**B**,**b**,**C**,**c**,**D**,**d**,**E**,**e**) after *H. diminuta* infection. Positive IHC reaction—brown reaction marked with arrows: red—epithelial cells of the villi and intestinal crypts, blue and white—cells in the connective tissue lining of the mucosa, green—intraepithelial lymphocytes. Lens magnification: ×20 (**A**–**E**, scale bar: 200 µm), ×40 (**a**–**e**, scale bar: 100 µm).

**Figure 8 ijerph-19-09753-f008:**
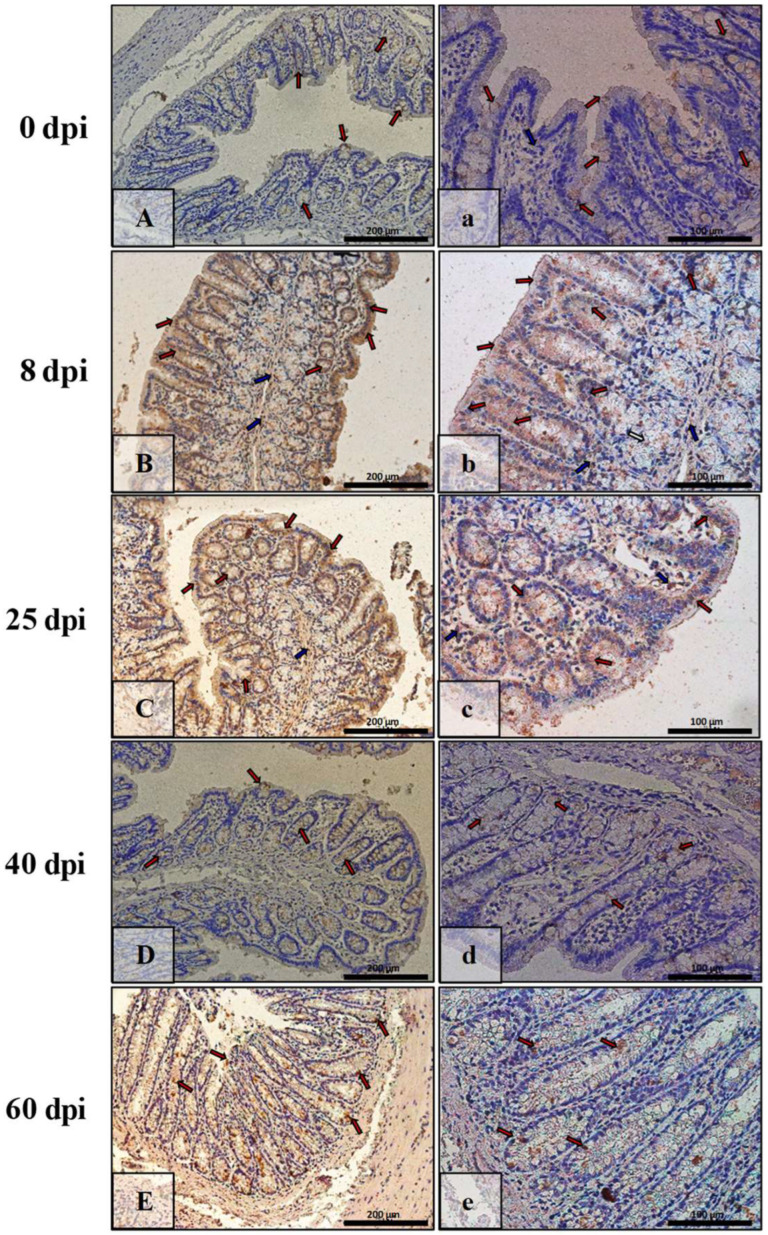
Representative microphotographs showing Bax immunoexpression in the colon of control animals (**A**,**a**) and on consecutive days (**B**,**b**,**C**,**c**,**D**,**d**,**E**,**e**) after *H. diminuta* infection. Positive IHC reaction—brown reaction indicated by arrows: red—epithelial cells of the intestinal villi and crypts, blue and white—cells in the connective tissue lining of the mucosa. Lens magnification: ×20 (**A**–**E**, scale bar: 200 µm), ×40 (**a**–**e**, scale bar: 100 µm).

**Figure 9 ijerph-19-09753-f009:**
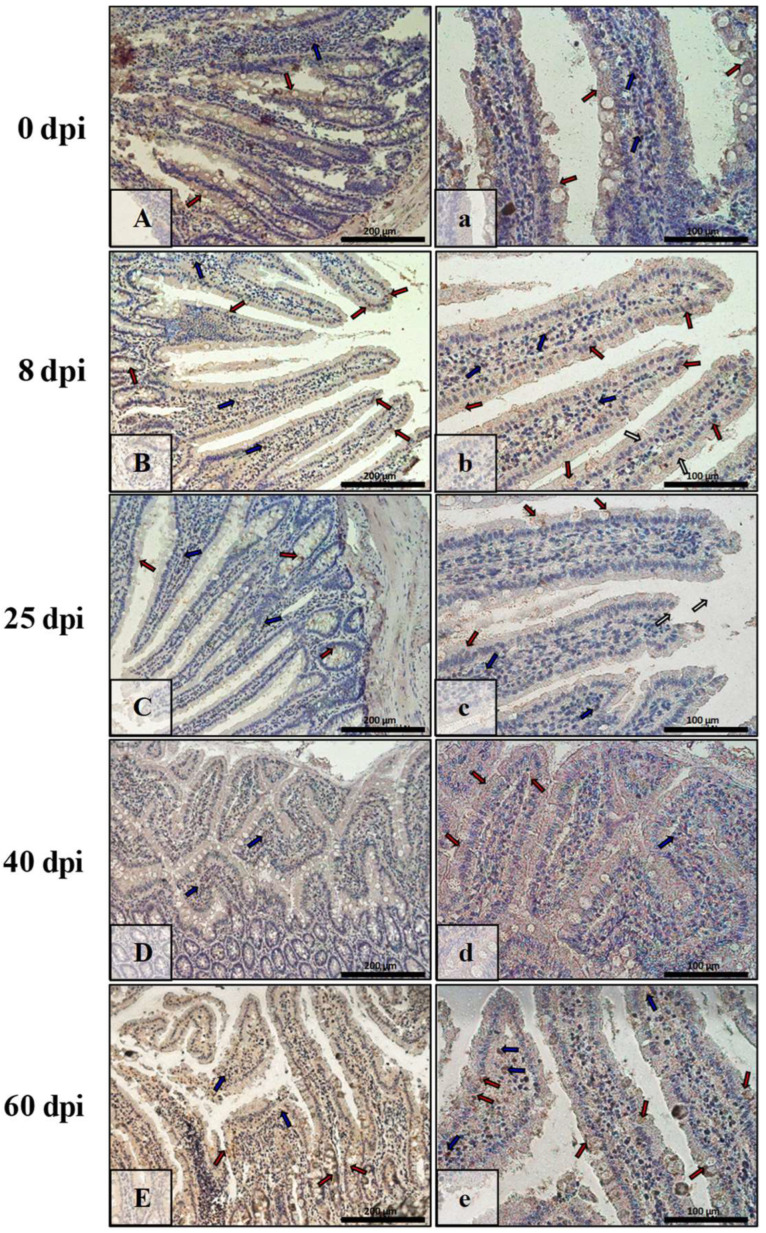
Representative microphotographs showing Bcl-2 immunoexpression in the small intestine of control animals (**A**,**a**) and on consecutive days (**B**,**b**,**C**,**c**,**D**,**d**,**E**,**e**) after *H. diminuta* infection. Positive IHC reaction—brown reaction marked with arrows: red—epithelial cells of the villi and intestinal crypts, blue and white—cells in the connective tissue lining of the mucosa. Lens magnification: ×20 (**A**–**E**, scale bar: 200 µm), ×40 (**a**–**e**, scale bar: 100 µm).

**Figure 10 ijerph-19-09753-f010:**
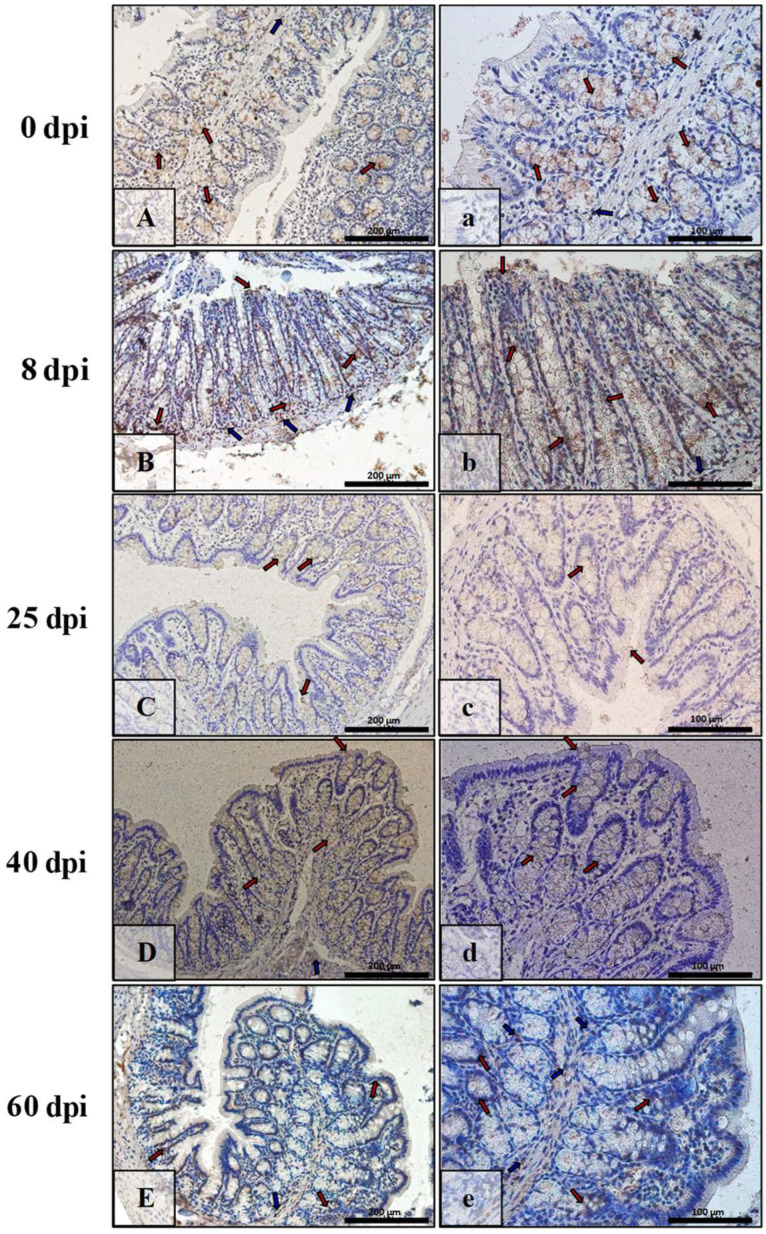
Representative microphotographs showing Bcl-2 immunoexpression in the colon of control animals (**A**,**a**) and on consecutive days (**B**,**b**,**C**,**c**,**D**,**d**,**E**,**e**) after *H. diminuta* infection. Positive IHC reaction—brown reaction indicated by arrows: red—epithelial cells of the intestinal villi and crypts, blue—cells in the connective tissue lining of the mucosa,. Lens magnification: ×20 (**A**–**E**, scale bar: 200 µm), ×40 (**a**–**e**, scale bar: 100 µm).

**Figure 11 ijerph-19-09753-f011:**
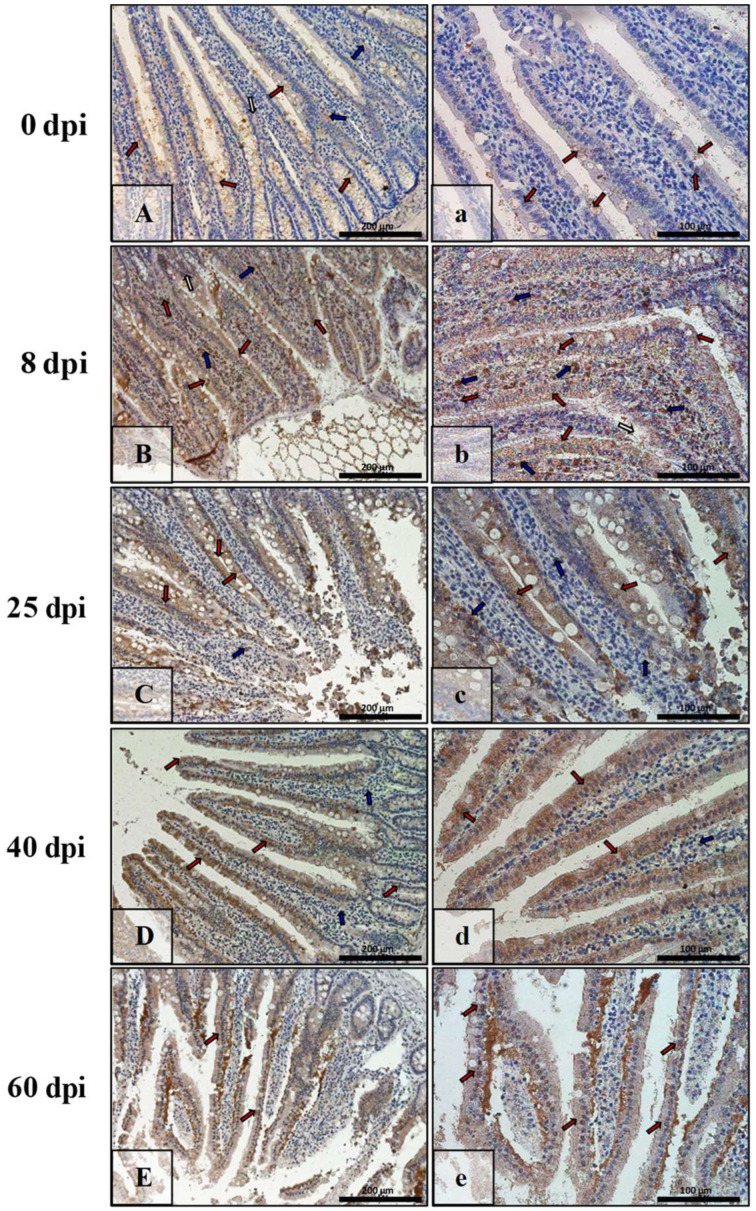
Representative microphotographs showing caspase-3 immunoexpression in the small intestine of control animals (**A**,**a**) and on subsequent days (**B**,**b**,**C**,**c**,**D**,**d**,**E**,**e**) after *H. diminuta* infection. Positive IHC reaction—brown reaction marked with arrows: red—epithelial cells of the villi and intestinal crypts, blue and white—cells in the connective tissue lining of the mucosa. Lens magnification: ×20 (**A**–**E**, scale bar: 200 µm), ×40 (**a**–**e**, scale bar: 100 µm).

**Figure 12 ijerph-19-09753-f012:**
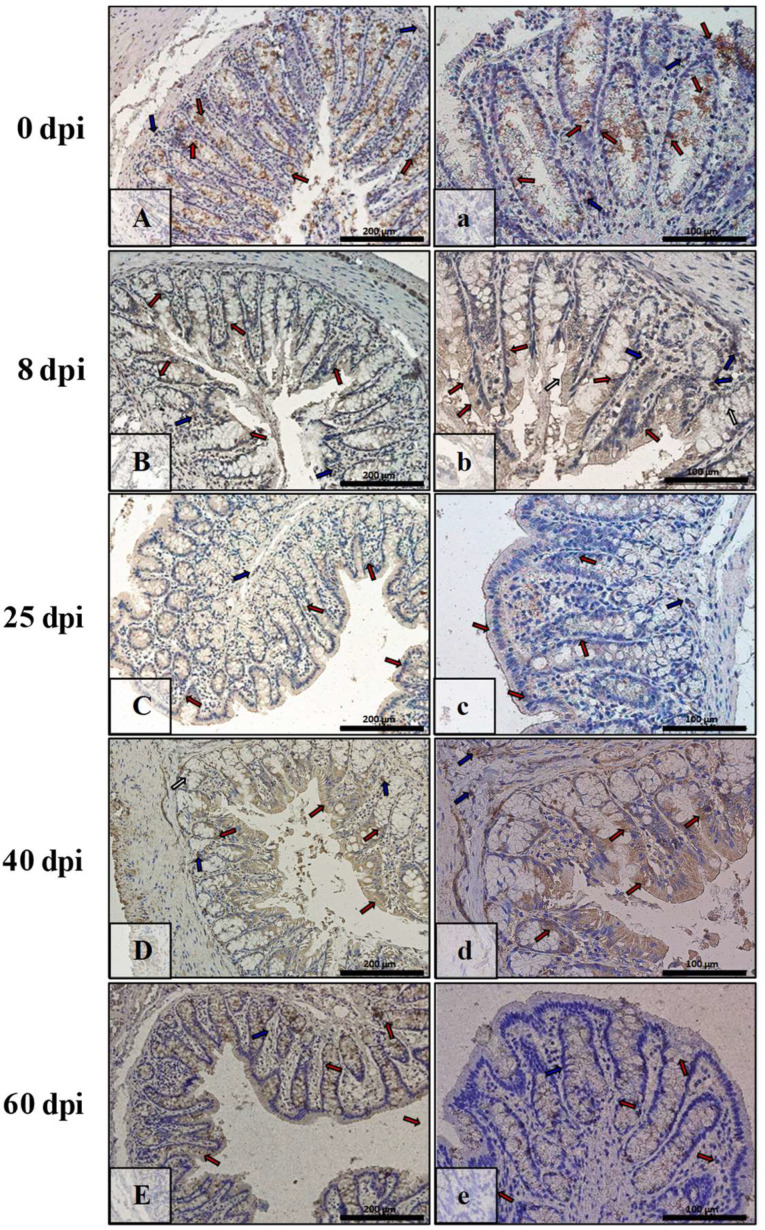
Representative microphotographs showing caspase-3 immunoexpression in the colon of control animals (**A**,**a**) and on consecutive days (**B**,**b**,**C**,**c**,**D**,**d**,**E**,**e**) after *H. diminuta* infection. Positive IHC reaction—brown reaction indicated by arrows: red—epithelial cells of the intestinal villi and crypts, blue and white—cells in the connective tissue lining of the mucosa,. Lens magnification: ×20 (**A**–**E**, scale bar: 200 µm), ×40 (**a**–**e**, scale bar: 100 µm).

**Figure 13 ijerph-19-09753-f013:**
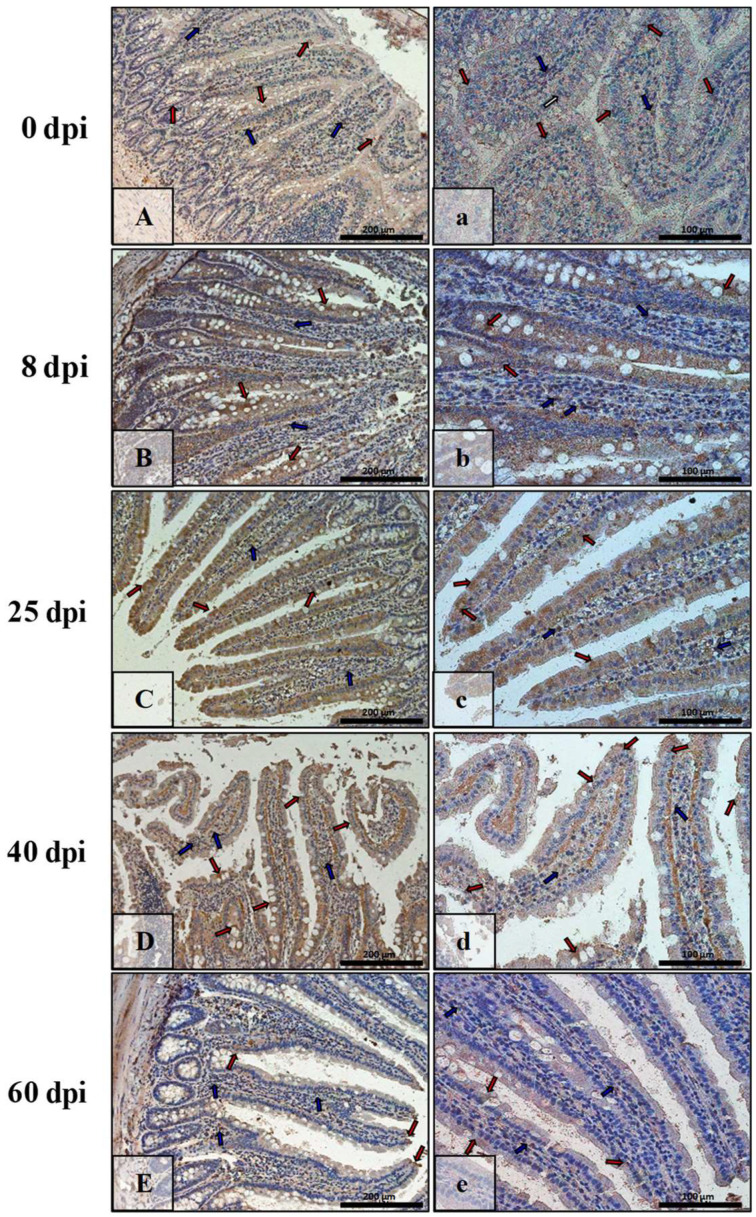
Representative microphotographs showing caspase-9 immunoexpression in the small intestine of control animals (**A**,**a**) and on subsequent days (**B**,**b**,**C**,**c**,**D**,**d**,**E**,**e**) after *H. diminuta* infection. Positive IHC reaction—brown reaction marked with arrows: red—epithelial cells of the villi and intestinal crypts, blue and white—cells in the connective tissue lining of the mucosa. Lens magnification: ×20 (**A**–**E**, scale bar: 200 µm), ×40 (**a**–**e**, scale bar: 100 µm).

**Figure 14 ijerph-19-09753-f014:**
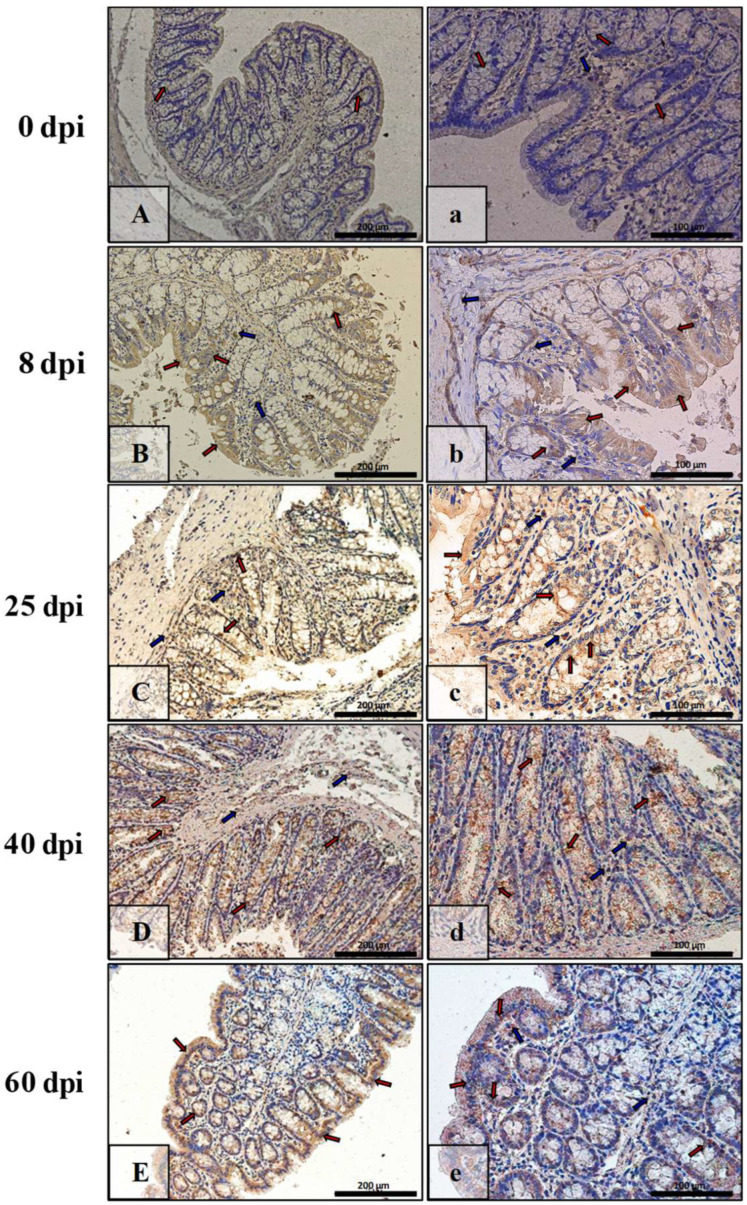
Representative microphotographs showing caspase-9 immunoexpression in the colon of control animals (**A**,**a**) and on consecutive days (**B**,**b**,**C**,**c**,**D**,**d**,**E**,**e**) after *H. diminuta* infection. Positive IHC reaction—brown reaction indicated by arrows: red—epithelial cells of the intestinal villi and crypts, blue—cells in the connective tissue lining of the mucosa. Lens magnification: ×20 (**A**–**E**, scale bar: 200 µm), ×40 (**a**–**e**, scale bar: 100 µm).

**Table 1 ijerph-19-09753-t001:** Primer sequences were used in RT-PCR.

	Forward	Reverse
**GAPDH**	*ATG ACT CTA CCC ACG GCA AG*	*CTG GAA GAT GGT GAT GGG TT*
**Bcl-2**	*ATC CAG GAT AAC GGA GGC TG*	*CAG GTA TGC ACC CAG AGT GA*
**Bax**	*GGG TGG TTG CCC TTT TCT ACT*	*AGT CCA GTG TCC AGC CCA TG*
**Cas-3**	*AAT TCA AGG GAC GGG TCA TG*	*TGA CAC AAT ACA CGG GAT CT*
**Cas-9**	*AGC CAG ATG CTG TCC CAT AC*	*CAG GAG ACA AAA CCT GGG AA*

**Table 2 ijerph-19-09753-t002:** Summary of the expression of Bax, Bcl-2, caspase-3, and caspase-9 in the control (0 dpi) and infected animals (8 dpi, 25 dpi, 40 dpi, 60 dpi) presented as intensity of immunostaining. Intensity of immunostaining scored as negative (−), very weakly positive (+/−), weakly positive (+), moderately positive (++), or strongly positive (+++).

Marker	Part of Intestine	0 dpi (Control)	8 dpi	25 dpi	40 dpi	60 dpi
**BAX**	small	+/−	+++	++	+	+++
	large	+/−	+++	+++	+	++
**Bcl-2**	small	+	+/−	+/−	++	++
	large	+	+	+/−	+/−	+/−
**Caspase-3**	small	+/−	+	++	+++	++
	large	+	++	+	++	+
**Caspase-9**	small	+/−	++	++	+	+/−
	large	+/−	+	+	++	++

## Data Availability

The data presented in this study are available on request from the corresponding author.

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
