# Peer review of "Hymenolepis diminuta Infection Affects Apoptosis in the Small and Large Intestine"

_ijerph, 2022, doi:10.3390/ijerph19159753_

Round 1
Reviewer 1 Report
The authors clarified the drawbacks I pointed previously regarding to time storage and melting curve in QT PCR. Nevertheless before publishing I suggest proof reading. Here are some examples:
"Figure 3: Reprseentative" - the space should be put after "3:”
The figures presenting QT PCR data end the same way: "Kruskal-Wallis test * p <0.05, ** p <0.01 vs 0 dpi". Comma should be put: "test, * p <0.05
Moreover authors used description of lens magnification and used e.g. "x20". The appropriate symbol "×" should be used instead of the letter "x"
Author Response
Review 1
The authors clarified the drawbacks I pointed previously regarding to time storage and melting curve in QT PCR. Nevertheless before publishing I suggest proof reading. Here are some examples:
"Figure 3: Reprseentative" - the space should be put after "3:”
Authors: corrected
The figures presenting QT PCR data end the same way: "Kruskal-Wallis test * p <0.05, ** p <0.01 vs 0 dpi". Comma should be put: "test, * p <0.05
Authors: corrected
Moreover authors used description of lens magnification and used e.g. "x20". The appropriate symbol "×" should be used instead of the letter "x"
Authors: Thank you very much for the in-depth evaluation of the article, a positive opinion, as well as the critical remarks which provide important indications that helped us improve the quality of the article.

Reviewer 2 Report
The submitted manuscript provides very interesting results dealing with the impact of Hymenolepis diminuta on apoptosis in small and large intestine in Wistar rats. This topic is very current and so far relatively understudied. The introduction is comprehensible and legibly written.
In the Material and methods, I lack information about the fact that a control of the success of the experimental infection was carried out (coprological examination 3 weeks after infection or confirmation of the presence of tapeworms in the intestine during dissection).
In the Results, I have a comment about data visualization in graphs. In my opinion, it is not possible to display the gene expression using bar plots (these are used to display counts). For this purpose it is appropriate to use box plots. Also, due to the non-normal distribution, it is more appropriate to display the median instead of the mean in the graph.
I have the following comments on the text (mostly it is about correcting typos):
line 55 – “ … Kosik-Bogackaet al. …” instead of “ … Kosik-Bogacka et al. …”
line 81 – “ … sinals …” instead of “ … signals …”
line 95 – “ … H. diminutamay…” instead of “ … H. diminuta may…”
line 283 –“ … thecolon …” instead of “ … the colon …”
line 306 – “ … 13%( not significant …” instead of “ … 13% (not significant …”
line 317 – “…there was an decreased of caspase 3…” should be “…there was an decrease of caspase 3…” or “…there was decreased caspase 3…”
line 318 – “At25 dpi …” instead of “At 25 dpi …”
line 331 – “… 8dpi …” instead of “ … 8 dpi …”
line 333 – “… was the lower …” should be ”… was the lowest …” or “… was lower …”
line 336 – “…(66%).At …” instead of “…(66%). At …” ; “ … 2%( not significant)....” instead of “ … 2% (not significant)….”
line 353 – “ … Baximmunoexpression … “ instead of “… Bax immunoexpression …”
line 395 – “… themucosa …” instead of “… the mucosa …”
line 417 – “ … withH. diminuta …” instead of “…with H. diminuta …”
line 539 – “G. intestinalistrophozoites …” instead of “G. intestinalis trophozoites…”
line 561-563 – Trichuris muris inhabits caecum and large intestine, so it is not so surprising that there is no effect on proliferation or apoptosis in the small intestine.
Author Response
Review 2
The submitted manuscript provides very interesting results dealing with the impact of Hymenolepis diminuta on apoptosis in small and large intestine in Wistar rats. This topic is very current and so far relatively understudied. The introduction is comprehensible and legibly written.
In the Material and methods, I lack information about the fact that a control of the success of the experimental infection was carried out (coprological examination 3 weeks after infection or confirmation of the presence of tapeworms in the intestine during dissection).
Authors: corrected
In the Results, I have a comment about data visualization in graphs. In my opinion, it is not possible to display the gene expression using bar plots (these are used to display counts). For this purpose it is appropriate to use box plots. Also, due to the non-normal distribution, it is more appropriate to display the median instead of the mean in the graph.
Authors: Gene expression was expressed as bar graphs. Due to the nature of the data, we believe that they represent the results better than box plots. For example, we present one box plot.
In addition, in publicly available articles, this form of presenting the results is acceptable, e.g.:
- D.I. Kosik-Bogacka, I. Baranowska-Bosiacka, A. Kolasa-Wołosiuk, N. Lanocha-Arendarczyk, I. Gutowska, J. Korbecki, H. Namięta, I. Rotter, The inflammatory effect of infection with Hymenolepis diminuta via the increased expression and activity of COX-1 and COX-2 in the rat jejunum and colon, Experimental Parasitology, Volume 169, 2016, Pages 69-76, ISSN 0014-4894,https://doi.org/10.1016/j.exppara.2016.07.009.
- Ma SL, Wu J, Zhu L, Chan RS, Wang X, Huang D, Tang NL, Woo J. Peripheral Blood T Cell Gene Expression Responses to Exercise and HMB in Sarcopenia. Nutrients. 2021 Jul 5;13(7):2313. doi: 10.3390/nu13072313. PMID: 34371826; PMCID: PMC8308783.
- Afshar Ebrahimi, F., Foroozanfard, F., Aghadavod, E. et al. The Effects of Magnesium and Zinc Co-Supplementation on Biomarkers of Inflammation and Oxidative Stress, and Gene Expression Related to Inflammation in Polycystic Ovary Syndrome: a Randomized Controlled Clinical Trial. Biol Trace Elem Res 184, 300–307 (2018). https://doi.org/10.1007/s12011-017-1198-5
I have the following comments on the text (mostly it is about correcting typos):
line 55 – “ … Kosik-Bogackaet al. …” instead of “ … Kosik-Bogacka et al. …”
line 81 – “ … sinals …” instead of “ … signals …”
line 95 – “ … H. diminutamay…” instead of “ … H. diminuta may…”
line 283 –“ … thecolon …” instead of “ … the colon …”
line 306 – “ … 13%( not significant …” instead of “ … 13% (not significant …”
line 317 – “…there was an decreased of caspase 3…” should be “…there was an decrease of caspase 3…” or “…there was decreased caspase 3…”
line 318 – “At25 dpi …” instead of “At 25 dpi …”
line 331 – “… 8dpi …” instead of “ … 8 dpi …”
line 333 – “… was the lower …” should be ”… was the lowest …” or “… was lower …”
line 336 – “…(66%).At …” instead of “…(66%). At …” ; “ … 2%( not significant)....” instead of “ … 2% (not significant)….”
line 353 – “ … Baximmunoexpression … “ instead of “… Bax immunoexpression …”
line 395 – “… themucosa …” instead of “… the mucosa …”
line 417 – “ … withH. diminuta …” instead of “…with H. diminuta …”
line 539 – “G. intestinalistrophozoites …” instead of “G. intestinalis trophozoites…”
line 561-563 – Trichuris muris inhabits caecum and large intestine, so it is not so surprising that there is no effect on proliferation or apoptosis in the small intestine.
Authors: all corrected
Authors: Thank you very much for the in-depth evaluation of the article, a positive opinion, as well as the critical remarks which provide important indications that helped us improve the quality of the article.

This manuscript is a resubmission of an earlier submission. The following is a list of the peer review reports and author responses from that submission.
Round 1
Reviewer 1 Report
Authors took a nice survey to shed new insights in the mechanisms/ outcomes of H. diminuta infection. The research concept is well thought. It contains investigation of pro- and anti -apoptotic factors on mRNA and a protein level. Consequently immunohistochemical analyses were performed. Nevertheless some information are missing and should be clarified. My main concern is time of sample storage. The material was collected few years ago.
- For validation and confirm the credibility of the results RNA quality should be confirmed using BioAnalyzer or at least non denaturing electrophoresis gel.
- Authors used primers spanning intron/exon for QT PCR analyses. That is wise choice. Nevertheless, the fact that the primers were designed to amplify the specific cDNA sequences does not mean they are specific. Experimental confirmation needs to be performed. I require information if the primers were used on template with genomic DNA. If they did and no product occurred the primers are fine. Otherwise they are nosuitable for further experiments.
- The QT PCR reaction does not finished with melting curve step what is necessary when SYBR GREEN kits.
- Authors depict some equation for the dCt calculation. The description is somewhat blurry. I suggest change it and just cite the method (Livak and Schmittingen)
- The sentence “i.e. the target site of the parasite” in abstact is a bit misfortunate.
- Line 43 authors: Authors cite the phenomenon taking place in Wistar rats. For Readers convenience and general knowledge some information regarding other rats would be desired.
- Line 56, 62: Authors describe that the infection can impact glutathione synthesis and cyclooxygenases. More detailed information is desired. How the infections impact those particles? Increased/ decrease expression (activity)
- Lines 46: 47. Authors write about decrease in transepithelial electrical potential difference. I am not expert in this field. However as far as I know TEER (transepithelial resistance ) is reversely correlated to the epithelium permeability with detailed mechanism described :
Cooke H.J. “Enteric Tears”: Chloride Secretion and Its Neural Regulation. News Physiol. Sci. 1998;13:269–274. doi: 10.1152/physiologyonline.1998.13.6.269.
9. Please explain exactly what transepithelial electrical potential difference is. Is it the same thing what TEER?
10. Lines 480- 489 (as other detailed descriptions of ant apoptotic action of investigated genes) should be moved to introduction section.
Author Response
|
Review 1 |
||||
Comments and Suggestions for Authors
The study was dedicated to the investigation of a mechanism of apoptosis in the rat intestinescaused by rat tapewormHymenolepisdiminuta. Fourproteins (markers of apoptosis) wereselected, and theirexpression was followedat mRNA and protein levelsatdifferenttimepoints. Unfortunately, the mainconclusion was a confirmation of the apoptosis, whichcould be anticipatedbased on the literature. The discussion part was morefocusedontopreviouspapers, rather the data obtained in the presentstudy. Authorsdid not discuss the process dynamics, did not explainedfluctuations in the expression of proteinsatdifferentlevelsduringinfection.
The data wereobtainedusing 10 yearsoldsamples. Whatis the influence of a long-term storageonto the samplesat RNA and protein levels?
Authors: We would like to thank the reviewer for this comment. Indeed, the date of approval for the research may suggest that the research has been going on for a long time. We would like to clarify that the samples on which the tests were performed were not stored for a long time. Immediately after the animal experiments and tissue sampling were performed, molecular tests were performed and the results were secured. The results were then successively used to explain the studied phenomena and prepare subsequent manuscripts.
RNA profiles do not reflect the respective protein ones (fig 1,2 vs fig 3-6).
Authors: We would like to thank the reviewer for this comment. Indeed, the results show that the host organism's response is different at the level of the gene and of the protein produced. This may indicate the influence of Hymenolepis diminuta infection not only on the activation of mechanisms influencing the transcription of the studied genes through the action of transcription factors, but also on post-transcriptional and post-translational mechanisms. This may lead to the induction of the observed changes in the tested proteins. However, this requires further research of these mechanisms.
Data presented on the figures 3-6 and table 1 are not in the agreement. For example, in fig 3B it’sshown that Bax protein decreased significantly at 25dpi, Chile according to table 1 it was marked as strongly positive. Or caspase 3 was at maximum levelat 0 dpi (fig 5B), while in table 1 the most intensivesignalswereat 8 and 40dpi.
The criteria of resultsevaluation in the manuscriptsometimesareunclear. For example, changes in immunoexpression of proteins in one ‘+’ (Table 1) discussed as ‘similarlevel’ (lines 396-398) and ‘increase/decrease’ (lines 409-411) simultaneously. Also, saying ‘decrease/increase of expression’ onto the samples with no statistical significance is speculation. Suchcasesshould be named ‘same level’.
Authors: It is assumed that immunohistochemical results are represented by a + or - sign.Intensity of immunostaining scored as negative (-), very weakly positive (+/-), weakly positive (+), moderately positive (++) or strongly positive (+++).
For example: Kupnicka, P.; Listos, J.; Tarnowski, M.; Kolasa-Wołosiuk, A.; Wąsik, A.; Łukomska, A.; Barczak, K.; Gutowska, I.; Chlubek, D.; Baranowska-Bosiacka, I. Fluoride Affects Dopamine Metabolism and Causes Changes in the Expression of Dopamine Receptors (D1R and D2R) in Chosen Brain Structures of Morphine-Dependent Rats. Int. J. Mol. Sci. 2020, 21, 2361. https://doi.org/10.3390/ijms21072361
The overalllevel of manuscriptpreparationislow: figure in polishlanguage (1B), wronglegends to the figures (whitearrows on the figurebecamegreen in the legend, fig 9), no error barsat the controlsamples (Fig. 1-6), numeroustypos in the text, organismnamemust be written in italic (in the title and abstractshould be given the fullname), statementscontroversial to the graphical materials.
Authors: figure in polishlanguage (1B)- correcteed. The error bars in the control groups are low and therefore may appear invisible.
Authors: Thank you very much for the in-depth evaluation of the article, a positive opinion, as well as the critical remarks which provide important indications that helped us improve the quality of the article.
Reviewer 2 Report
The study was dedicated to the investigation of a mechanism of apoptosis in the rat intestines caused by rat tapeworm Hymenolepis diminuta. Four proteins (markers of apoptosis) were selected, and their expression was followed at mRNA and protein levels at different time points. Unfortunately, the main conclusion was a confirmation of the apoptosis, which could be anticipated based on the literature. The discussion part was more focused onto previous papers, rather the data obtained in the present study. Authors did not discuss the process dynamics, did not explained fluctuations in the expression of proteins at different levels during infection.
The data were obtained using 10 years old samples. What is the influence of a long-term storage onto the samples at RNA and protein levels?
RNA profiles do not reflect the respective protein ones (fig 1,2 vs fig 3-6).
Data presented on the figures 3-6 and table 1 are not in the agreement. For example, in fig 3B it’s shown that Bax protein decreased significantly at 25dpi, while according to table 1 it was marked as strongly positive. Or caspase 3 was at maximum level at 0 dpi (fig 5B), while in table 1 the most intensive signals were at 8 and 40dpi.
The criteria of results evaluation in the manuscript sometimes are unclear. For example, changes in immunoexpression of proteins in one ‘+’ (Table 1) discussed as ‘similar level’ (lines 396-398) and ‘increase/decrease’ (lines 409-411) simultaneously. Also, saying ‘decrease/increase of expression’ onto the samples with no statistical significance is speculation. Such cases should be named ‘same level’.
The overall level of manuscript preparation is low: figure in polish language (1B), wrong legends to the figures (white arrows on the figure became green in the legend, fig 9), no error bars at the control samples (Fig. 1-6), numerous typos in the text, organism name must be written in italic (in the title and abstract should be given the full name), statements controversial to the graphical materials.
Author Response
Review 2
Authorstook a nice survey to shednewinsights in the mechanisms/ outcomes of H. diminuta infection. The researchconceptiswellthought. It containsinvestigation of pro- and anti -apoptoticfactors on mRNA and a protein level. Consequentlyimmunohistochemicalanalyseswereperformed. Neverthelesssomeinformationare missing and should be clarified. My mainconcernistime of samplestorage. The material was collectedfewyears ago.
- For validation and confirm the credibility of the results RNA qualityshould be confirmedusingBioAnalyzeroratleast non denaturingelectrophoresisgel.
- Authorsusedprimersspanning intron/exon for QT PCR analyses. Thatiswise choice. Nevertheless, the factthat the primersweredesigned to amplify the specificcDNAsequencesdoes not meantheyarespecific. Experimentalconfirmationneeds to be performed. I requireinformationif the primerswereused on template with genomic DNA. Iftheydid and no productoccurred the primersare fine. Otherwisetheyarenosuitable for furtherexperiments.
Authors: We would like to thank the reviewer for this comment. Indeed, the date of approval for the research may suggest that the research has been going on for a long time. We would like to clarify that the samples on which the tests were performed were not stored for a long time. Immediately after the animal experiments and tissue sampling were performed, molecular tests were performed and the results were secured. The results were then successively used to explain the studied phenomena and prepare subsequent manuscripts.
- The QT PCR reactiondoes not finished with meltingcurve step whatisnecessarywhen SYBR GREEN kits.
- Authorsdepictsomeequation for the dCtcalculation. The descriptionissomewhatblurry. I suggestchangeit and justcite the method (Livak and Schmittingen)
Authors: corrected
- The sentence “i.e. the target site of the parasite” in abstactis a bit misfortunate.
Authors: Corrected
- Line 43 authors: Authorscite the phenomenontaking place in Wistar rats. For Readers convenience and generalknowledgesomeinformationregardingotherratswould be desired.
Authors: Corrected
- Line 56, 62: Authorsdescribethat the infectioncanimpactglutathionesynthesis and cyclooxygenases. Moredetailedinformationisdesired. How the infectionsimpactthoseparticles? Increased/ decreaseexpression (activity)
Autors: ,, In conclusion, we found that experimental hymenolepidosis is
accompanied by increased LPO, changes in anti-oxidant enzyme
activity and altered GSH level in the gastrointestinal tract, which
may indicate a decrease in the efficiency of intestinal protection
against oxidative stress induced by the presence of the parasite.
Imbalance between oxidant and anti-oxidant processes can play
a major role in pathology associated with hymenolepidosis, ex-
pressed in previously described changes in the transport of ions
in the epithelium (Kosik-Bogacka et al., 2010, 2011) and morpho-
logical changes (Fal and Czaplicka, 1991; Martin and Holland,
1984) of the gastrointestinal tract of rats infected with these
worms. Additionally, as the level of oxidative stress markers and
the activity of anti-oxidant enzymes in the tapeworm H. diminuta
during the infection of rats is also high (Twarowska et al., 2010),
it indicates the formation of an adaptive parasite-host relationship”
(Kosik-Bogacka D., Baranowska-Bosiacka I., Noceń I., Jakubowska K., Chlubek D. Hymenolepis diminuta: Activity of anti-oxidant enzymes in defferent parts of rat gastrointestinal tract. Exp. Parasitol. 2011;128:265–271. doi: 10.1016/j.exppara.2011.02.026.)
- Lines 46: 47. Authorswriteaboutdecrease in transepithelialelectricalpotentialdifference. I am not expert in this field. However as far as I know TEER (transepithelialresistance ) isreverselycorrelated to the epitheliumpermeability with detailedmechanismdescribed :
Cooke H.J. “EntericTears”: ChlorideSecretion and ItsNeuralRegulation. News Physiol. Sci. 1998;13:269–274. doi: 10.1152/physiologyonline.1998.13.6.269.
- Pleaseexplainexactlywhattransepithelialelectricalpotentialdifferenceis. Isit the same thingwhat TEER?
Authors: 2.4. Changes in Ion Transport in the Host’s Digestive Tract
,,Mechanisms of glucose transport, ion transport such as Na+, K+ and Cl−, and water in the digestive tract, are very complex [68]. In the small intestine, the transport of water takes place by means of simple diffusion, and the absorption of electrolytes is performed, inter alia, via active transport. The basolateral membrane of the intestine is responsible for the absorption of Na+ via active transport, and the upper part of the small intestine is involved in the transport of Cl− by passive diffusion [69]. As demonstrated, ion transport processes may be disturbed at the moment of stimulation of the immune system due to an infection with intestinal parasites [70,71]. The key role in these mechanisms is played by the lymphatic tissue found in the mucosa and submucosa of the gastrointestinal tract [72]. The effect on intestinal transport and motor activity is also exerted by mediators of inflammation: mast cells, macrophages, neutrophils and eosinophils released due to the presence of the pathogens [70]. An example of this is mast cell secretion of histamine, which affects the secretion of chloride ions and inhibits the absorption of NaCl in the gastrointestinal tract of the rat [73].
Since H. diminuta lives in the small intestine of the host, it interferes with the digestion and absorption in the host [35,70]. Research shows that H. diminuta can affect the absorption process by sequestering 3′,5-cyclic guanosine monophosphate (cGMP), which connects to the mucous membrane of the intestine via cGMP receptors, while altering intestinal motility and slowing down intestinal processes [74].
The transport of ions through the epithelium can be carried out by transcellular and extracellular pathways. Transcellular transport is targeted (dependent on the transport systems in cell membranes) and active (requires energy). The basic measure of the electrogenic transport of the ions is the transepithelial electrical potential (PD). The value of this parameter depends, inter alia, on the transport of sodium and chlorine ions through ion channels located in epithelial cells. In contrast, extracellular transport is bi-directional and passive. It occurs through gap junctions in the basolateral and in the apical part via tight junctions between epithelial cells [75]. The integrity of tight junctions and the degree of permeability of a given tissue for ions is determined by the measurement of transepithelial electrical resistance (R).” (from Kapczuk, Patrycja et al. “Selected Molecular Mechanisms Involved in the Parasite⁻Host System Hymenolepis diminuta⁻Rattus norvegicus.” International journal of molecular sciences vol. 19,8 2435. 17 Aug. 2018, doi:10.3390/ijms19082435)
- Lines 480- 489 (as otherdetaileddescriptions of ant apoptoticaction of investigatedgenes) should be moved to introductionsection.
Authors: Corrected
Authors: Thank you very much for the in-depth evaluation of the article, a positive opinion, as well as the critical remarks which provide important indications that helped us improve the quality of the article.
Reviewer 3 Report
The paper submitted by Kapczuk P. et al entitled "H. diminuta infection affects apoptosis in the small and large intestine" examined the role of H. diminuta in the apoptosis process, that occurs in intestine. The question underlying the analysis - whether H. diminuta infection can affect apoptosis in the rat intestine, is explained in deep, materials and methods section is extremely detailed, thus the study is potentially well reproducible. The subject addressed in this article is worthy of investigation. The information presented is new. Results section is detalied and clear and figures helped in understanding the obtained results. Discussion is well explained. The conclusions are supported by the data and the organization of the manuscript is appropriate. Figures are correct with one mistake - in Fig.1 graph B is in Polish. It is probably small mistake. In general the manuscript is appropriate for the journal.
Author Response
Review 3
The papersubmitted by Kapczuk P. et al entitled "H. diminutainfectionaffectsapoptosis in the small and large intestine" examined the role of H. diminuta in the apoptosisprocess, thatoccurs in intestine. The questionunderlying the analysis - whether H. diminutainfectioncanaffectapoptosis in the rat intestine, isexplained in deep, materials and methodssectionisextremelydetailed, thus the studyispotentiallywellreproducible. The subjectaddressed in thisarticleisworthy of investigation. The informationpresentedisnew. Resultssectionisdetalied and clear and figureshelped in understanding the obtainedresults. Discussioniswellexplained. The conclusionsaresupported by the data and the organization of the manuscriptisappropriate. Figuresarecorrect with one mistake - in Fig.1 graph B is in Polish. It isprobably small mistake. In general the manuscriptisappropriate for the journal.
Authors:Thank you very much for the in-depth substantive evaluation of the article and a positive opinion.
Round 2
Reviewer 2 Report
None corrections were made to results/discussion/conclusion sections. All points raised in my first revision still stand on